# Stable topical application of antimicrobials using plumbing rings in an *ex vivo* porcine corneal infection model

**Daniel M. Foulkes**[1,2]*, **Keri Mclean**[1,2], **Tarunima Sharma**[2], **David G. Fernig**[1], **Stephen B. Kaye**[1]

**1** Department of Eye and Vision Science, Institute of Life Course and Medical Sciences, University of Liverpool, Liverpool, United Kingdom, **2** Department of Biochemistry, Institute of Systems, Molecular and Integrative Biology, University of Liverpool, Liverpool, United Kingdom

* Daniel.foulkes@liverpool.ac.uk

## Abstract

Microbial keratitis (MK) is a substantial cause of clinical blindness worldwide. *Pseudomonas aeruginosa* is an opportunistic Gram-negative bacterium and is the leading cause of MK. Infection models are vital tools in understanding host-pathogen interactions and the development of novel therapies. As well as ethical and practical advantages, *ex vivo* infection models enable researchers to study host-pathogen interactions with greater accuracy and physiological relevance compared to traditional cell culture systems. The versatility of porcine corneal *ex vivo* models have been employed to study various pathogens (for example *Staphylococcus aureus* and *Acanthamoeba*) and has enabled innovation of novel MK therapies. Here, we describe an improved porcine corneal *ex vivo* protocol, which uses plumbing rings and medical adhesive to circumvent several distinct limitations and challenges. The application of a 10 mm plumbing ring to the center of the cornea allows localized inoculation of pathogens of interest, maintaining them at the site of infection, rather than running the risk of "run off" of topically added aqueous solutions. The second important advantage is that topically applied therapeutic agents can be properly maintained on the cornea within the plumbing ring reservoir, allowing more accurate study of antimicrobial effects. In this contextualized protocol, we infected porcine corneas with the *P. aeruginosa* strain PA103 with topical treatments of moxifloxacin. PA103 colony-forming unit (CFU) quantification, spectrophotometric measurement of corneal opacity, and histological analysis of stromal edema using hematoxylin and eosin staining were employed to assess infection over 48 hours. Moxifloxacin treatment demonstrated a dose-dependent reduction in infection and corneal damage. These findings have contributed to the development of an improved and standardized *ex vivo* infection model for evaluating therapeutic interventions, potentially supporting clinical translation to alleviate the burden of microbial keratitis.

## 1. Introduction

Corneal opacification is the sixth leading cause of global blindness, accounting for 3.2% (0.5%-7.2%) of 36 million cases. Microbial keratitis (MK) is the main cause of corneal opacification.

**Data availability statement:** All relevant data are within the manuscript and its Supporting information files. This submission contains the minimal data set.

**Funding:** Funded by The Fight for Sight Carmen Butler-Charteris Charitable Trust. Project: 5175/5176. The funders had no role in study design, data collection and analysis, decision to publish, or preparation of the manuscript.

**Competing interests:** The authors have declared that no competing interests exist.

The corneal epithelium maintains a barrier to invading microorganisms, however, infection may occur when this barrier is disrupted by trauma, contact lens wear or surgery [1]. MK normally presents with a painful epithelial defect with associated signs of corneal stromal inflammation and ulceration. There is often a localized corneal opacity beneath and surrounding the ulcer, accompanied by an anterior uveitis, fibrinous exudate and/or a hypopyon. Infection may progress to a descemetocele and perforation and occasionally MK can cause whole eye loss, which is more frequently observed in older patients [1].

*Pseudomonas aeruginosa* is an opportunistic gram-negative pathogenic bacterium and accounts for 25% of MK cases. *P. aeruginosa* harbors a diverse array of virulence factors, among which Type III secretion system (T3SS) effectors and secreted proteases play pivotal roles in the pathogenesis of microbial MK. T3SS is a fundamental driver of *P. aeruginosa* pathogenicity, of which the two main effectors are Exotoxin U (ExoU) [2,3] and Exotoxin S (ExoS) [4,5]. The ExoU phospholipase causes rapid cell lysis and its expression is associated with increased MK disease severity with poor clinical outcomes [6]. ExoS modulates the cytoskeleton of host cells by ADP-ribosylating specific host targets such as Ras and Rho proteins, facilitating bacterial invasion and intracellular survival of *P. aeruginosa* in chronic infections [7]. Several key secreted proteases of *P. aeruginosa*, including elastases (LasA and LasB), protease IV and *P. aeruginosa* small protease (PASP), target distinct host molecules and are considered critical to MK pathogenesis by driving tissue damage, degrading extracellular matrix (ECM) components, and subverting the host immune response [8–10].

Commissioned by the UK parliament, it is predicted in O'Neill's Review on Antimicrobial Resistance that by 2050, deaths resulting from infections by antibiotic resistant bacteria will surpass deaths from cancer [11]. The alarming efficiency in which bacteria have developed resistance against our existing therapeutic arsenal of antimicrobial agents, means that there is an urgent need to develop new therapies. Due to their accessibility and with ethical considerations (reducing dependency on extensive *in vivo* studies), *ex vivo* models are becoming increasing popular tools to interrogate mechanisms of infection and assess novel therapeutics. Porcine eyes are inexpensive (our lab is charged £2 per eye) and are available from local abattoirs. From histological and immunohistochemical characterization, it has been demonstrated that, despite the apparent absence of a Bowman's layer, the porcine corneal structure and molecular markers closely resembles that of a human being's, despite possessing greater thickness (1100-1497 µM for porcine and 500-550 µM for human) [12,13]. This increase in thickness, however, means that porcine corneas possess a greater available area for pathophysiological investigations. Beyond MK, the *ex vivo* cornea is a valuable testbed to assess infection mechanisms and test novel therapeutics, as it is an accessible, robust and transparent organ, which may also give researchers the opportunity to translate their findings to other infections sites.

In recent years, there have been various developments and optimizations in the use of *ex vivo* porcine eye models for the testing of new therapies. Kennedy *et al.*, (2020) developed an *ex vivo* porcine corneal *P. aeruginosa* infection model to test the efficacy of patented poly-epsilon-lysine peptide hydrogels in the treatment of MK [14]. Whole porcine corneas were dissected with surgical scissors and scalpel, placed epithelial side down into sterile bijou tube lids with agarose dissolved in Dulbecco's modified eagles' medium (DMEM) then added to produce a cornea with agarose support. When placed epithelial side up in a 6-well plate, pathogen and/or therapeutic agent can then be added to the corneal dome. It is important to note that to achieve consistent infection, the epithelial barrier must be first damaged. Some studies opt for the debriding of epithelial cells by application of 70% ethanol [14,15], while others report mechanical trauma with a scalpel or needle tip (30G) to be effective [16,17]. In this study, we employed ethanol debridement as we found this to be less technically challenging than needle scratching and more reproducible.

With both bacteria and therapeutic agent being topically added to the cornea, there is significant risk that aqueous solutions run from the center of the cornea into the 6-well plate, precluding experimentation. To circumvent this challenge, Okurowska *et al.*, (2020) developed a protocol that employs custom made glass molds that fit around the whole cornea with a 10 mm opening at the top in which to add aqueous solutions [16]. Although these bespoke glass molds are innovative, their production requires a skilled glass blower within a specialized facility.

In the present study, we adapted the protocol of Kennedy *et al.*, (2020) to include the attachment of plumbing rings (readily available from hardware stores), with medical adhesive, to the surface of the cornea. The plumbing rings act as a reservoir whereby an aqueous volume of up to 150 µL can be added and maintained in a centrally localized position on dissected porcine corneas. The application of plumbing rings with medical adhesive to create an apical reservoir on *ex vivo* corneas does not require extensive technical skill. Moreover, the reagents used are inexpensive and readily available to any lab.

As proof of concept for our plumbing ring application method, we infected *ex vivo* porcine corneas with the cytotoxic, ExoU producing, strain of *P. aeruginosa*, PA103, and observed effects of infection after 48h. Moxifloxacin is a fluroquinolone antibiotic that is often used to treat *P. aeruginosa* MK. Therefore, we topically applied varying concentrations of moxifloxacin to infected porcine corneas to test the efficacy of an established MK therapeutic in the developed *ex vivo* model. Measurements included, imaging of corneas, quantification of opacity, deduction of PA103 colony forming units (CFU) in homogenized corneas, and histological analysis of sectioned corneas by hematoxylin and eosin (H&E) staining. We describe an improved standardized *ex vivo* infection model which can be used to better evaluate MK treatments.

## 2. Materials and methods

The protocol described in this peer-reviewed article is published on protocols.io, https://dx.doi.org/10.17504/protocols.io.eq2ly6nkmgx9/v1 and is included for printing as supporting information file 1 with this article.

### 2.1. Materials

The following materials and reagents were required for this study: whole pig eyes obtained the morning of slaughter (CS Morphet's abattoir, Widnes, UK), surgical scissors, forceps and scalpel (size 10, Fisher scientific), phosphate-buffered saline (PBS), 10% (v/v) iodinated povidone (Ecolab), ethanol, 10 mm Plumbsure Rubber O rings (B&Q, Bidston, UK), LIQUIBAND 0.8 g Exceed topical skin adhesive (Advanced Medical Solutions, Netherlands), Luria–Bertani (LB) broth, LB agar (Melford), UltraPure Agarose (Thermo Fisher), DMEM with 10% (v/v) fetal bovine serum (Labtech), 12-well plates, moxifloxacin (Merck) and H&E staining kit (Abcam).

### 2.2. Preparation of bacterial inoculum

The full step-by-step guide for the developed *ex vivo* porcine cornea infection model can be found in S1. The *P. aeruginosa* strain PA103 was cultured in 100 mL of LB broth, from a glycerol stock, overnight at 37ºC in a shaker incubator. 2 mL of the PA103 overnight culture was then added to 20 mL of fresh LB broth and expanded to an $OD_{600}$ of 0.8. Of this subculture, 1 mL was taken and centrifuged for 5 minutes at 5000 **g**. The supernatant was discarded, and the bacterial pellet resuspended in 10 mL of PBS, yielding a PA103 suspension at a concentration of $10^7$ CFU/mL.

### 2.3. Preparation of porcine corneas

The eyes of large white landrace pigs were removed the morning of slaughter and transferred to PBS. In a laminar flow hood, using forceps to hold the eyes in place, a small incision was made at the edge of the sclera, releasing ocular pressure. Surgical scissors were then used to cut around the cornea, whilst maintaining ~ 2.5 mm of sclera outside the circumference of the cornea in a full circle. The resulting corneoscleral button was separated from the uveal tissue using two sets of forceps. In a sterile 10 cm dish, corneas were subsequently incubated in 10% iodinated povidone for 2 minutes for sterilisation. Corneas were then washed 3 times in PBS. In another 10 cm dish, one side of 10 mm plumbing rings were immersed onto LIQUIBAND topical skin adhesive. Forceps were then used to compress all edges of the ring to the center of the cornea, ensuring a tight seal.

Corneas were placed epithelial side down. UltraPure Agarose (0.5% [w/v]) was dissolved in DMEM in a 65.5 °C water bath, cooled to approximately 37 °C and then pipetted onto the endothelial side of corneas to fill the cavity. Corneas with solidified agarose supports were transferred into 12-well plates, epithelial side up.

To induce a controlled epithelial wound on the cornea, 10 µL 70% ethanol (v/v in PBS) was applied to the center of the corneal surface for 10 seconds. Following ethanol exposure, the epithelial layer was debrided using the edge of a scalpel around the treated area to simulate mechanical injury. In the 12-well plate, the corneas were rinsed three times with PBS to ensure complete removal of ethanol and debris. Finally, 1 mL of DMEM was added to each well, covering the outer edge of the corneal epithelium without submerging the plumbing ring.

### 2.4. Inoculation of corneas and topical application of MK treatment

For each cornea that was infected, 10 µL of a $10^7$ CFU/mL suspension of PA103 in PBS was added to Eppendorf tubes with the concentrations of moxifloxacin indicated in the figure legends. The final volume of each suspension was made up to 150 µL with PBS, which was then added to the plumbing ring reservoir at the centre of the cornea. This equated to an addition of 1 x$10^5$ CFU of PA103 inoculated onto each cornea. Corneas, in 12-well plates, were transferred to a 37 °C incubator in 5% $CO_2$ (v/v) for 48 h.

### 2.5. Determination of CFU within corneas

After 48 h infection, corneas were gently separated from the agarose supports and plumbing rings and then washed in PBS. Corneas were then transferred into 500 µL PBS and homogenized using a Qiagen Tissue ruptor (Qiagen). Suspended bacteria were serially diluted in PBS ($10^{-1}$-$10^{-6}$) with 10 µL plated onto LB agar plates followed by overnight incubation at 37°C. After incubation, colonies were counted on plates with bacterial load calculated by multiplying the counted colonies by dilution factor and the plated volume.

### 2.6. Imaging and spectrophotometric analysis of corneal opacity

After 48 hours of infection, corneas were removed from agarose supports and plumbing rings and then rinsed in PBS. To assess corneal opacity visually, corneas were imaged using a BioRad ChemiDoc system, capturing optical density data. For quantitative analysis, corneas were transferred to a 12-well plate, and opacity was measured using a Hidex spectrophotometer by evaluating transmitted light permeability at 400 nm.

### 2.7. Histological analysis of porcine corneas

Corneas were incubated in 4% paraformaldehyde (PFA) for 24 h and then processed using a Leica ASP300 tissue processor. Tissues were fixed in formalin, dehydrated through a graded

series of ethanol, cleared in xylene, and then embedded in paraffin wax. Five μm sections of paraffin-embedded corneal tissues were cut using a microtome and mounted onto glass slides.

For histological analysis, the tissue sections were stained with H&E using a staining kit (Abcam), following the manufacturer's protocol. Once stained, the sections were imaged using a VENTANA DP 200 slide scanner (Roche). The open-source software QuPath was used to view and analyse images for morphological evaluation of corneal tissue and to assess the extent of infection and tissue damage. ImageJ was utilised to quantify stromal swelling in corneal sections by measuring the percentage of white space within the stroma. Images were converted to 8-bit grayscale, and the stroma was selected as the region of interest (ROI), excluding the epithelium and Descemet's membrane. Threshold values were adjusted to a range of 230–255 to isolate white pixels representing stromal space. The percentage area of white pixels within the ROI was then calculated and analysed using GraphPad Prism.

### 2.8. Statistical analysis

All results presented are from three independent experiments from porcine eyes obtained on three separate occasions. Results are presented as mean $\pm$ standard deviation (SD). Where appropriate, one-way ANOVA with multiple comparisons was performed in Prism (*<0.05, **<0.01, ***<0.001, ****<0.0001).

### 2.9. Ethics statement

Porcine eyes used in this study were obtained from CS. Morphet and Sons abattoir (Widnes, UK) as by-products of commercial slaughter practices. No animals were specifically killed for the purpose of this research. The procurement and use of these tissues complied with ethical guidelines and applicable laws governing the use of animal-derived materials. Ethical approval for the use of these materials was deemed not necessary, as no live animals were involved in the study.

## 3. Results and discussion

We tested the efficacy of moxifloxacin to reduce PA103 infection of porcine corneas, *ex vivo*. In our developed protocol, PA103 inoculum and aqueous moxifloxacin was maintained on the cornea with use of a plumbing ring reservoir, employing the preparatory method described in Fig 1. Downstream analysis consisted of corneal CFU count, corneal opacity and histology analysis.

### 3.1. CFU analysis to determine efficacy of moxifloxacin in PA103 infected porcine corneas

PA103-infected corneas, in the absence of moxifloxacin, exhibited a bacterial expansion from an initial inoculum of 5 $\log_{10}$ CFU to 10.5 $\pm$ 0.6 $\log_{10}$ CFU after 48 h of infection (Fig 2). This level of *P. aeruginosa* growth aligns with findings from previous studies [16]. Prior research has demonstrated that *P. aeruginosa* reaches a maximum growth of approximately $10^9$-$10^{10}$ CFU within 24 hours post-infection, with extended incubation (48 hours) showing no significant further increase in bacterial load. This plateau in growth after prolonged incubation is likely due to nutrient depletion, which limits further bacterial expansion.

By administering increasing concentrations of moxifloxacin to PA103-infected corneas, we observed a dose-dependent reduction in viable CFU recovered from homogenized corneas (Fig 2). All acquired CFU results for the dose escalation of moxifloxacin treatment can be found in S1 Table. Our previous *in vitro* studies determined the minimal inhibitory concentration (MIC) of moxifloxacin for PA103 growth to be 4 μM [18]. Although

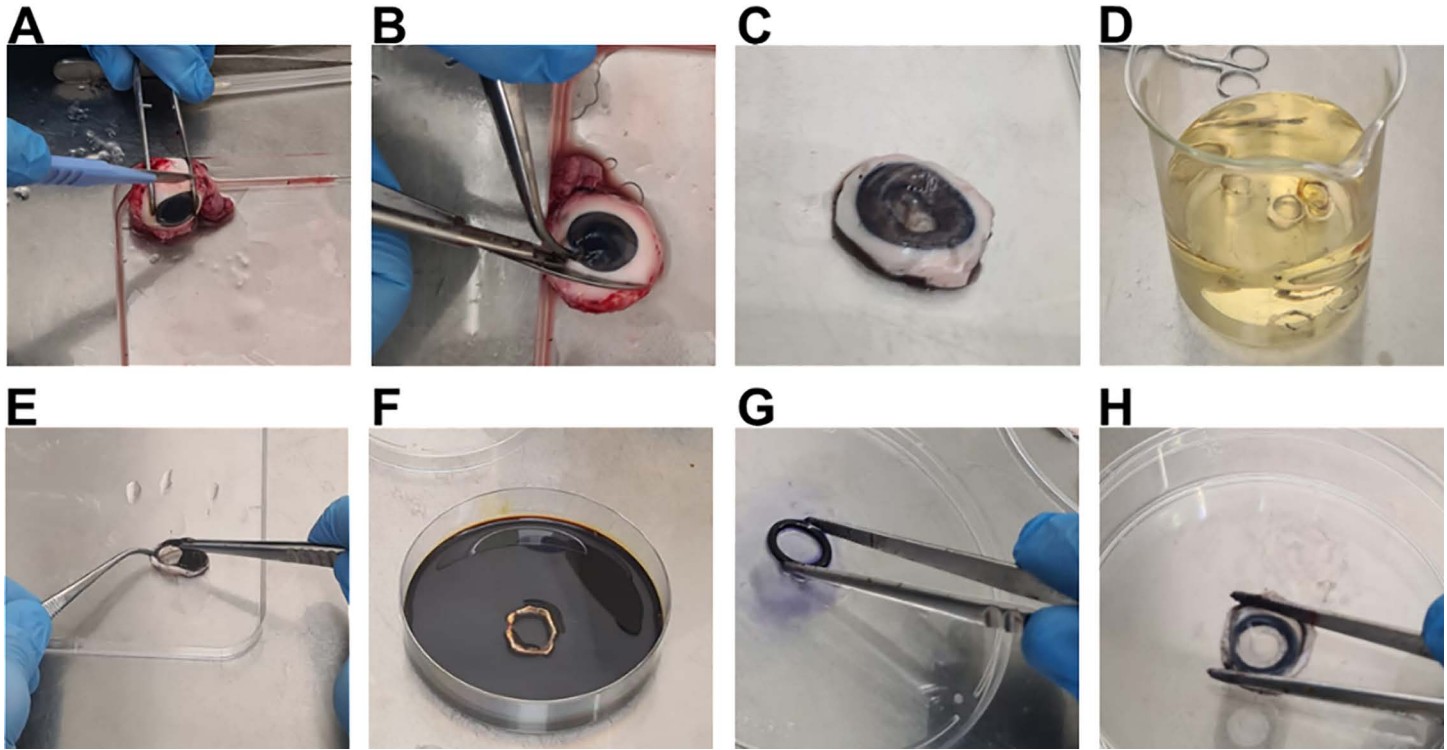

**Fig 1. Application of plumbing ring during porcine cornea preparation.** The key steps in the preparation of the porcine cornea and the assembly of the plumbing ring reservoir are shown. (A) A scalpel was used to make an incision into the sclera. (B) Scissors were used to cut around the cornea to yield (C) a corneal scleral button. (D) Two pairs of forceps were used to remove the uveal tissue. (E) The corneas were immersed in 10% (v/v) iodinated povidone for 2 minutes and then washed 3 times in PBS (F). (G) LIQUIBAND adhesive was applied to a sterile 10 cm petri dish, and a 10 mm plumbing ring was immersed in this. (H) Using forceps, the epithelial side of the porcine corneas was pressed onto the adhesive-coated side of the plumbing ring, ensuring an even seal.

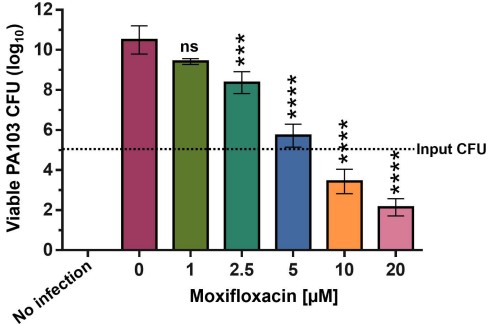

**Fig 2. PA103 CFU detection in *ex vivo* porcine corneas after 48h infection and moxifloxacin treatment.** *Ex vivo* porcine corneas were infected with $1 \times 10^5$ CFU PA103 for 48 h in the indicated concentrations of moxifloxacin. Whole corneas were homogenized, diluted in PBS followed by plating, overnight incubation at 37 °C prior to CFU counting. Results plotted are mean ± SD from 3 independent experiments. For statistical analysis, one-way ANOVA with multiple comparisons (to 0 µM moxifloxacin) was performed in Prism (*<0.05, **<0.01, ***<0.001, ****<0.0001).

significantly lower than non-treated controls, at 5 µM moxifloxacin, there was a slight increase in detectable PA103 CFU compared to the initial inoculum. Even at 5 times the *in vitro* MIC (20 µM), while a significant reduction was observed, 2.1 ± 0.3 $\log_{10}$ CFU of viable PA103 persisted. This is consistent with recent findings where the related fluoroquinolone,

ciprofloxacin, failed to completely eliminate *P. aeruginosa* from *ex vivo* porcine corneas, even at concentrations above the MIC [19]. The concentrations of moxifloxacin used in this study are substantially lower than those typically employed in clinical settings, potentially explaining the incomplete clearance of PA103 from the corneas. Additionally, factors such as antibiotic bioavailability and penetration in *ex vivo* porcine corneas, as well as the absence of fully operational host immune defenses, may also contribute to the observed persistence of bacteria.

### 3.2. Measurement of corneal opacity to determine moxifloxacin effectiveness

Topical solutions of moxifloxacin added to PA103-infected *ex vivo* porcine corneas resulted in a dose-dependent reduction in corneal opacity after 48 hours, as measured by optical density at 400 nm (Fig 3). In the absence of moxifloxacin, a significant increase in opacity was observed, consistent with severe infection and corneal damage caused by *P. aeruginosa*. Treatment with moxifloxacin at 1, 2.5, and 5 µM led to modest decreases in opacity, suggesting limited efficacy in mitigating infection-related corneal clouding at these concentrations. However, 10 µM moxifloxacin significantly reduced opacity, while 20 µM restored corneal transparency to levels comparable to uninfected controls. Corneal opacity measurements for each experiment are displayed in S2 Table, which may guide the reader of expected results using this protocol.

Despite the restoration of corneal clarity at 20 µM moxifloxacin, viable *P. aeruginosa* were still detected, as evidenced by CFU counts (Fig 2). The extent of bacterial load may not directly translate to a proportional increase in corneal opacity. We hypothesise that the degree of opacity is primarily driven by bacterial virulence factors rather than the bacterial load itself and that these factors modulate the structure of the epithelium and endothelium, leading to a loss of corneal transparency. At 20 µM, moxifloxacin likely suppressed bacterial virulence and toxin production, reducing tissue damage and clouding despite the persistence of bacteria.

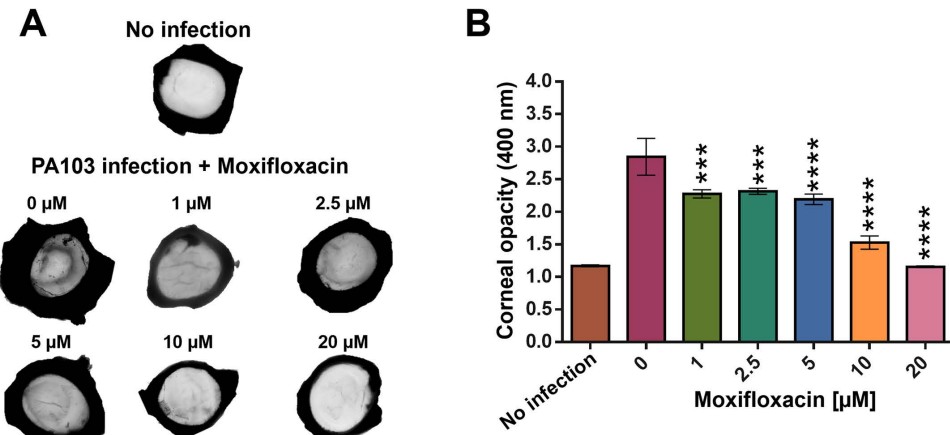

**Fig 3. Opacity measurement to determine moxifloxacin efficacy in PA103 infected *ex vivo* porcine corneas.** The plumbing rings reservoir on the *ex vivo* porcine corneas was filled with 150 µL PBS containing $10^5$ CFU PA103 and varying concentrations of moxifloxacin. After 48h incubation the corneas were imaged and their opacity measured. (A) Images of the corneas were acquired with a Bio-Rad Chemidoc. (B) The opacity of the corneas was determined by measuring their optical absorbance at 400 nm in a 12-well plate using a spectrophotometer. Results of 3 independent experiments were plotted and one-way ANOVA with multiple comparisons (to 0 µM moxifloxacin) was performed in Prism (*<0.05, **<0.01, ***<0.001, ****<0.0001).

Previously, we demonstrated, *in vitro*, that moxifloxacin concentrations greater than 4 µM reduced PcrV expression [18], a critical component of the T3SS. In these experiments, moxifloxacin at higher concentrations may have reduced both virulence and bacterial load to a level that, while not entirely eradicating the infection, was insufficient to induce significant tissue degradation associated with corneal opacity. This suggests that while moxifloxacin at 20 µM was able to preserve corneal integrity, complete bacterial clearance may require higher concentrations, additional treatments or a functioning immune system to fully eliminate the pathogen.

### 3.3. Histology to evaluate PA103 infection after moxifloxacin treatment

Using histology, with H&E staining, it is possible to assess the structural changes and erosion that occurs to the cornea during *P. aeruginosa* infection. Typically, during infection, cellular damage, thinning and ulceration (erosion) can be observed in the epithelium (Fig 4A, black arrows). In the stroma, edema can be observed with increased space between collagen fibrils (Fig 4, white arrows), which can be quantified (Fig 4B). It is also possible to employ gram staining to observe quantity and penetration of infection [15].

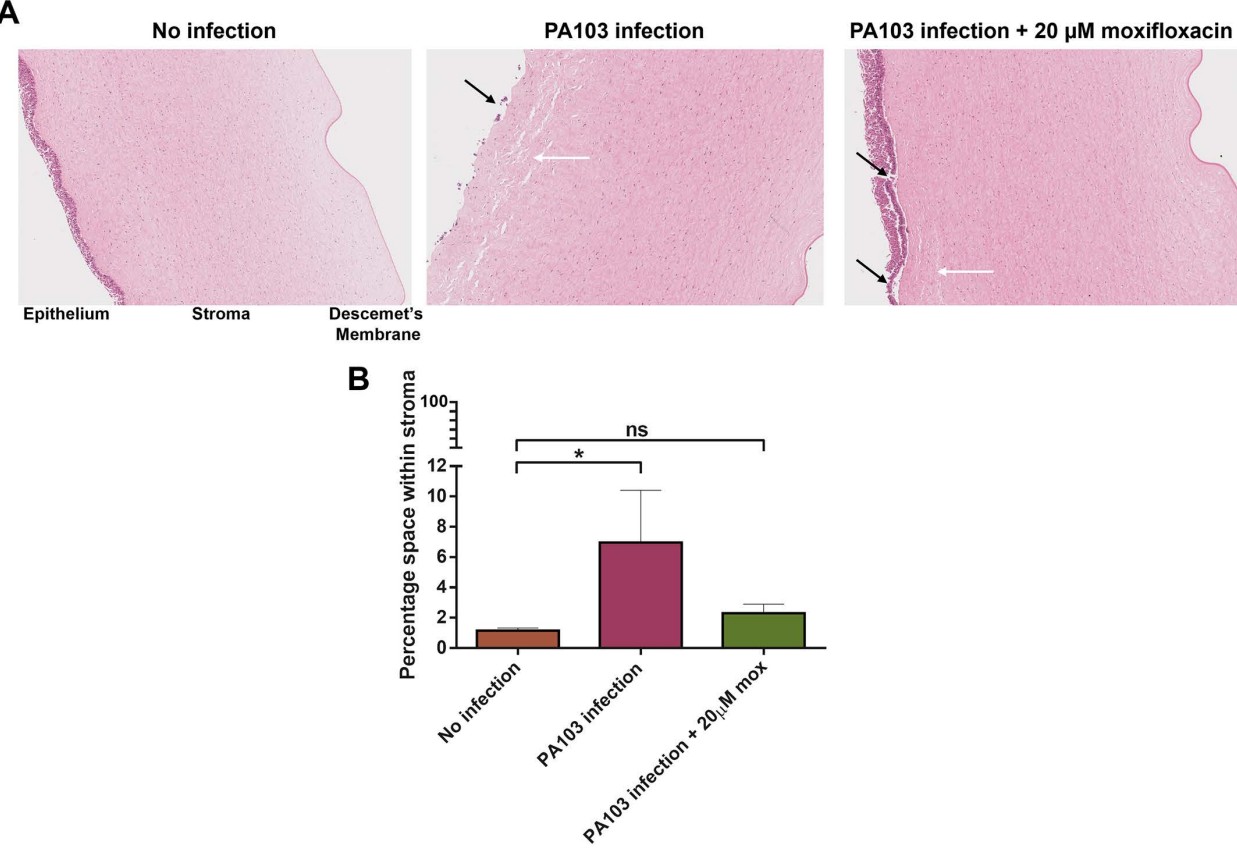

**Fig 4. Histological analysis of PA103 infected porcine *ex vivo* corneas.** *Ex vivo* porcine corneas were infected with 1 x 10⁵ CFU of PA103 for 48h with and without 20 moxifloxacin. After parafilm embedding and sectioning, slides were stained with H&E. (A) Stained corneal sections were imaged using a Ventana DP 200 slide scanner. Black arrows indicate epithelial ulceration and white arrows stromal edema. (B) The percentage of space within the corneal stroma in response to PA103 infection was assessed using ImageJ. The stroma of the corneal sections was designated as the region of interest (ROI) and converted to 8-bit grayscale for analysis. A threshold range of 230–255 was applied to identify and quantify white space within the stroma. The resulting data were plotted using GraphPad Prism, and statistical significance was evaluated using one-way ANOVA with multiple comparisons (*p < 0.05, **p < 0.01, ***p < 0.001, ****p < 0.0001).

Without infection, regular and undamaged epithelial layers with intact stroma could be observed (Fig 4A). After 48h PA103 infection, we observed complete epithelial erosion and extensive stromal edema (Fig 4A). Quantification of the white space within the stroma allowed us to assess the extent of stromal swelling (Fig 4B). In the absence of infection, the stroma exhibited minimal swelling, with 1.2% ± 0.1 white space. In contrast, infection with PA103 resulted in significant stromal edema, with 7.0% ± 2.8 white space. Treatment with 20 µM moxifloxacin largely preserved the integrity of the corneal epithelium (Fig 4A) and reduced stromal swelling to 2.3% ± 0.5 (Fig 4B). In all instances, the Descemet's membrane remained undamaged, which would otherwise indicate extensive infection and complete corneal breakdown. We note that it is not usually possible, due to processing limitations, to retain the squamous endothelium, which resides beyond the Descemet's membrane. Images used to quantify stromal swelling are shown in S1 Fig with analysed data in S3 Table.

## 4. Limitations

### 4.1. Porcine cornea physiology considerations

While a porcine cornea shares many anatomical and physiological similarities with that of a human being [12,13], certain differences, such as the greater thickness of a porcine cornea, may influence the progression of infections and the permeability of therapeutic agents. This difference could lead to altered permeation rates and retention times for drugs in porcine corneas, particularly for hydrophilic drugs, which face more resistance crossing the denser stroma [20]. This discrepancy could impact the direct translation of findings to human ocular infections. Moreover, unlike human corneas, porcine corneas lack a Bowman's membrane, a structure that serves as a protective barrier and contributes to resistance against injury and infection. However, porcine corneas have evolved to maintain transparency and structural integrity, potentially compensating for the absence of the Bowman's membrane through additional epithelial cell layers. Although the *ex vivo* porcine model offers a valuable platform for studying infection mechanisms and testing novel treatments, it is crucial to validate findings in *in vivo* models. This step is essential to assess therapeutic efficacy in a more complex biological environment before considering clinical applications.

### 4.2. Limitations of the *ex vivo* environment

Despite its advantages, the porcine eye *ex vivo* model has distinct limitations. The lack of a fully functional immune system in this model excludes innate and adaptive host immune responses, which play a critical role in infection control and healing *in vivo*. Thus, although resident macrophages and low numbers of neutrophils are present in the cornea prior to infection [21], the lack of a circulatory system in *ex vivo* models limit its ability to fully replicate the complex interactions between bacteria and the host's immune response. This limitation likely contributes to the incomplete eradication of PA103 by moxifloxacin after 48 hours (Fig 2). Additionally, the restricted availability of nutrients in *ex vivo* systems may alter bacterial behavior and treatment efficacy, further complicating the model's ability to fully mimic *in vivo* infections.

The absence of tear film, which contains antimicrobial peptides, enzymes, immunoglobulins and surfactants, removes an essential component of the eye's natural defense. Further developments to the porcine *ex vivo* cornea model could consist of the incorporation of artificial tear solutions, containing components such as electrolytes (NaCl, KCl, $MgCl_2$), lysozyme, lipids, and mucins [22,23]. *In vitro* these are used to mimic natural tear fluid and are often used in drug permeability studies. Advanced *in vitro* models, such as tear film lipid layer and microfluidic tear film models might be incorporated into *ex vivo* experimental design to study drug dissolution, and pathogen interaction with the ocular surface.

To further enhance the physiological relevance of the *ex vivo* porcine corneal model, incorporating co-culture systems with immune cells could provide valuable insights into inter-tissue and cell-cell interactions. For example, introducing neutrophils to the basal side of the cornea could facilitate the study of inflammation and neutrophil infiltration, particularly in transepithelial migration assays. Similarly, advancements could include integrating microfluidic or perfusion platforms that enable dynamic flow and the controlled delivery of immune cells, cytokines, and antimicrobial factors. Peng et al. (2022) developed an innovative *ex vivo* model of human corneal rim perfusion organ culture, whereby corneal rims are glued to 3D-printed perfusion plates [24]. Adaptation of systems such as this would allow researchers to simulate physiologically relevant conditions, such as immune cell-driven or infection-induced cytokine gradients, thereby offering a more comprehensive understanding of immune responses and their impact on corneal health and pathology.

### 4.3. Experimental design considerations

Certain experimental limitations must be carefully considered during experimental design. Previous studies have shown that even a low initial inoculum of *Pseudomonas aeruginosa* (cytotoxic strain PA14), as few as 215 CFU, is sufficient to establish infection in *ex vivo* porcine corneas [19]. Remarkably, the bacterial load consistently reached a maximum of approximately $1 \times 10^9$ CFU after 18 hours of infection, regardless of the initial inoculum size (ranging from 215 to $1 \times 10^7$ CFU). While inoculating with such a low CFU count can enhance the dynamic range for observing bacterial load reductions in response to treatments, it also introduces potential variability due to the challenges of accurately mixing and pipetting at these low concentrations. Therefore, careful consideration of the initial inoculum size is essential, balancing the need for sensitivity in detecting treatment effects with the desire to minimize experimental variation.

In our experience, endothelial cells are not consistently recovered for histological analysis following dissection. The corneal endothelium is a single layer of cells that is crucial for maintaining corneal hydration by regulating fluid exchange, but these cells are delicate and easily damaged during handling and processing, which might present limitations when studying highly penetrant infections. While the *ex vivo* porcine eye model allows for the evaluation of the integrity of the Descemet's membrane, the inability to reliably analyze the endothelium hinders comprehensive investigation of deeper corneal infections. In cases involving corneal perforation or complete corneal melting, histological analysis is further compromised, preventing detailed insight into the fate of the endothelial layer. As a result, this model may not fully capture the extent of endothelial damage in severe or advanced infections, necessitating complementary approaches to fully understand these processes.

### 4.4. Choice of plumbing ring size

The selection of a 10 mm plumbing ring was critical to the experimental design as it provided an adequate and standardized surface area for the application of inoculum and topical treatments, thereby maximizing the reliability and consistency of analysis outputs. This size proved to be universally compatible with all porcine corneas tested, ensuring that the rings could be securely affixed without impinging on the sclera. Larger plumbing rings, by contrast, occasionally extended beyond the corneal boundary onto the sclera, particularly in cases where variations in eye size were encountered due to differences in specimens obtained from the abattoir. This observation underscores the importance of tailoring the experimental setup to accommodate anatomical variability in biological specimens.

Smaller plumbing rings may present an alternative when a more localized area of infection is required. For instance, a reduced initial infection area could be advantageous for studying pathogens like herpes simplex virus, which often induces localized lesions. By adjusting the size of the ring, researchers can modify the infection site to better suit the specific pathophysiological characteristics of the disease being studied. This flexibility in experimental design highlights the utility of the plumbing ring approach for a wide range of infectious disease models. Overall, the 10 mm size provides a balance between providing sufficient area for comprehensive treatment analysis and maintaining compatibility across diverse specimens, making it an optimal choice for this particular study.

## 5. Conclusions

In this study, we describe our advances in using the porcine eye *ex vivo* model as a simple, robust, reproducible, and cost-effective method for investigating ocular infections. A key challenge we addressed was the tendency for bacterial inoculum and topically applied treatments to run off the corneal epithelium, preventing consistent infection and treatment delivery. By incorporating plumbing rings, we were able to localize both the bacterial inoculum and therapeutic compounds, covering a substantial portion of the corneal surface while maintaining consistent exposure. Prior to this modification, it was difficult to reliably achieve reproducible *P. aeruginosa* infections in porcine corneas when applying aqueous therapeutics. While the isolated application of small, stable volumes (5-10 µL) directly onto the center of the cornea is sufficient for testing solid agents like hydrogels or contact lenses [14,15], larger aqueous volumes often could not be retained on the corneal surface. This led to uncertainty regarding the effective concentration of the applied treatment and how much compound remained in contact with the cornea. Using moxifloxacin as a standard treatment, in this contextualized protocol we demonstrate the effectiveness of moxifloxacin to reduce corneal opacity, epithelial ulceration and stromal edema, highlighting the use of the developed *ex vivo* porcine eye model to assess topical therapeutics for bacterial infection. With this, we present a highly accessible and improved *ex vivo* infection model, which may lead to better clinical translation of prospective treatments to reduce the burden of MK.

## Supporting information

**S1 Protocol. Describes the full step-by-step protocol developed for the study.**
(PDF)

**S1 Table. Corneal CFU after moxifloxacin treatments.** PA103 CFU detection in *ex vivo* porcine corneas after 48h infection and moxifloxacin treatment. *Ex vivo* porcine corneas were infected with 1 x $10^5$ CFU PA103 for 48 h in the indicated concentrations of moxifloxacin. Whole corneas were homogenized, diluted in PBS followed by plating, overnight incubation at 37 ºC prior to CFU counting. All data collected are presented in the Table along with mean ± SD from 3 independent experiments.
(TIF)

**S2 Table. Corneal opacity after moxifloxacin treatments.** Opacity measurement to determine moxifloxacin efficacy in PA103 infected *ex vivo* porcine corneas. The plumbing rings reservoir on the *ex vivo* porcine corneas was filled with 150 µL PBS containing $10^5$ CFU PA103 and varying concentrations of moxifloxacin. After 48h incubation the opacity of the corneas was measured using a spectrophotometer. All data collected are presented in the Table along with mean ± SD from 3 independent experiments.
(TIF)

**S1 Fig. Images of H&E stained corneal sections used for the quantification of stromal swelling.** The images used for the quantification of stromal edema from Fig 4B in the main manuscript. *Ex vivo* porcine corneas were infected with1 x 10⁵ CFU of PA103 for 48h with and without 20 µM moxifloxacin. Corneas were sectioned, mounted and stained with H&E. A Ventana DP 200 slide scanner was used to capture cross sections of stained corneal sections.) Images of H&E stained corneal sections used for the quantification of stromal swelling (Fig 4B in the main manuscript). *Ex vivo* porcine corneas were infected with1 x 10⁵ CFU of PA103 for 48h with and without 20 µM moxifloxacin. Corneas were sectioned, mounted and stained with H&E. A Ventana DP 200 slide scanner was used to capture cross sections of stained corneal sections.
(TIF)

**S3 Table. Quantification of Stromal White Space in Corneal Sections Following PA103 Infection.** The percentage of space within the corneal stroma in response to PA103 infection was assessed using ImageJ. The stroma of the corneal sections was designated as the region of interest (ROI) and converted to 8-bit grayscale for analysis. A threshold range of 230–255 was applied to identify and quantify white space within the stroma. The resulting data are presented in the table, which is derived from the 3 images for each condition shown in S1 Fig.
(TIF)

## Author contributions

**Conceptualization:** Daniel M. Foulkes, David G. Fernig, Stephen B. Kaye.

**Formal analysis:** Daniel M. Foulkes.

**Funding acquisition:** Stephen B. Kaye.

**Investigation:** Daniel M. Foulkes, Tarunima Sharma.

**Methodology:** Daniel M. Foulkes, Keri Mclean.

**Project administration:** Daniel M. Foulkes, David G. Fernig, Stephen B. Kaye.

**Supervision:** Daniel M. Foulkes, David G. Fernig, Stephen B. Kaye.

**Validation:** Daniel M. Foulkes.

**Writing – original draft:** Daniel M. Foulkes.

**Writing – review & editing:** Daniel M. Foulkes, Keri Mclean, David G. Fernig, Stephen B. Kaye.

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
