## [Decision Letter · Decision Letter 0]

7 Jan 2025

PONE-D-24-57134Stable topical application of antimicrobials using plumbing rings in an *ex vivo*  porcine corneal infection modelPLOS ONE

Dear Dr. Foulkes,

Thank you for submitting your manuscript to PLOS ONE. After careful consideration, we feel that it has merit but does not fully meet PLOS ONE’s publication criteria as it currently stands. Therefore, we invite you to submit a revised version of the manuscript that addresses the points raised during the review process.

We look forward to receiving your revised manuscript.

Kind regards,

Sanhita Roy, Ph.D.

Academic Editor

PLOS ONE

Journal Requirements:

Funded by The Fight for Sight Carmen Butler-Charteris Charitable Trust. Project: 5175/5176

4. In the online submission form, you indicated that The data underlying the results presented in the study are available on request from either Daniel Foulkes (daniel.foulkes@liverpool.ac.uk) or Stephen Kaye (sbkaye@liverpool.ac.uk).

Additional Editor Comments :

Please address the comments made by the reviewers.

Reviewers' comments:

Reviewer's Responses to Questions

**Comments to the Author**

1. Does the manuscript report a protocol which is of utility to the research community and adds value to the published literature?

Reviewer #1: Yes

Reviewer #2: Yes

2. Has the protocol been described in sufficient detail?

To answer this question, please click the link to protocols.io in the Materials and Methods section of the manuscript (if a link has been provided) or consult the step-by-step protocol in the Supporting Information files.

The step-by-step protocol should contain sufficient detail for another researcher to be able to reproduce all experiments and analyses.

Reviewer #1: Yes

Reviewer #2: Yes

3. Does the protocol describe a validated method?

Reviewer #1: Yes

Reviewer #2: Yes

4. If the manuscript contains new data, have the authors made this data fully available?

Reviewer #1: Yes

Reviewer #2: Yes

**5. Is the article presented in an intelligible fashion and written in standard English?**

Reviewer #1: Yes

Reviewer #2: Yes

6. Review Comments to the Author

Reviewer #1: This manuscript describes an improved protocol for studying microbial keratitis (MK) using an ex vivo porcine corneal infection model. The authors adapted an existing protocol by incorporating plumbing rings to maintain localized application of pathogens and therapeutic agents on the corneal surface. The study focuses on Pseudomonas aeruginosa infection and evaluates the efficacy of moxifloxacin in reducing bacterial load, corneal opacity, and tissue damage. The findings demonstrate the practicality and reliability of this model for testing antimicrobial treatments, emphasizing its potential applications in ocular research.

Major point:

- The introduction effectively highlights the significance of microbial keratitis and the advantages of ex vivo models. However, the rationale for selecting plumbing rings over other methods (e.g., glass molds) could be elaborated.

- Consider briefly addressing the clinical relevance of the findings, such as potential applications for human ocular infections.

- The CFU analysis in Figure 2 provides compelling evidence of moxifloxacin’s efficacy. Could the authors clarify whether the dose-dependent reduction was statistically significant across all tested concentrations?

- Figure 3 shows the restoration of corneal clarity at 20 µM moxifloxacin. The discussion notes incomplete bacterial clearance despite reduced opacity. Can the authors speculate further on the threshold of bacterial load required to induce opacity?

- The histological analysis (Figure 4) effectively demonstrates structural preservation with moxifloxacin treatment. However, additional quantitative metrics, such as epithelial thickness or stromal edema, could strengthen the findings.

- The preparation of bacterial inoculum and porcine corneas is described in detail. However, the use of 70% ethanol for epithelial debridement might alter corneal properties. Have the authors considered alternative methods?

- The choice of 10 mm plumbing rings is practical, but information on how ring size or adhesive type affects results would be beneficial.

- The methods lack specific information about the statistical analysis software and parameters. Please provide these details.

- The discussion addresses key limitations, including the absence of immune responses in ex vivo models. Could the authors propose strategies for overcoming these limitations, such as co-culture systems with immune components?

- The statement about reduced PcrV expression with moxifloxacin is intriguing but not directly tested in this study. Consider elaborating on how this mechanism aligns with the observed outcomes.

- The discussion could benefit from a stronger conclusion regarding the translational potential of the findings for human therapy.

Minor opinions:

- Abstract: Line 12 “interaction more accurately than” – consider rephrasing for clarity.

- Introduction: Line 43, “P. aeruginosa possesses an array of virulence factors” – consider specifying which are most relevant to MK.

- Methods: Line 127, “a concentration of 107 CFU/mL” – use superscripts for scientific notation.

- Figures: Ensure figure legends are self-explanatory without requiring text from the main body for context.

- Formatting: Verify consistent use of italicization for species names throughout the manuscript.

Reviewer #2: Microbial keratitis is a signifcant problem worldwide that can lead to sight loss. The bacterium Pseudomonas aeruginosa is a major pathogen in this regard and increases in antibiotic resistance are exacerbating the problem of bacterial keratitis. Further improved models of micribial keratitis are undoubteldy required to help combat this disease. The current submission describes an ex vivo whole porcine corneal model for studying bacterial keratitis that offers some improvements on previous porcine corneal models described by Okurowska et al 2020 and Kent et al 2020. The authors successfully monitor the effects of moxifloxacin on P. aeruginosa infection to assess the model. The system is simple, cost-effective and ethically acceptable, and the authors correctly acknowledge the limitations of the porcine model. The paper is well written and clear and represents an advance in the models that are currently availabe for investigating this important disease.

7. PLOS authors have the option to publish the peer review history of their article (what does this mean? ). If published, this will include your full peer review and any attached files.

**Do you want your identity to be public for this peer review?** For information about this choice, including consent withdrawal, please see our Privacy Policy .

Reviewer #1: No

Reviewer #2: No

---

## [Author Response · Author response to Decision Letter 1]

3 Feb 2025

Responses to Reviewer 1

We sincerely thank the reviewer for their thoughtful comments and valuable feedback. We hope that we have addressed all comments satisfactorily. Should the reviewer have further suggestions or areas they believe warrant elaboration, we would be happy to address them.

Major points:

1. The introduction effectively highlights the significance of microbial keratitis and the advantages of ex vivo models. However, the rationale for selecting plumbing rings over other methods (e.g., glass molds) could be elaborated.

The authors thank the reviewer for this observation and agree that further highlighting the advantages of using plumbing rings over glass moulds outlines them as a strong alternative. To make the logistical limitations more clear for the use of glass moulds (requiring a skilled glass blower and specialised facilities), we have introduced a separate paragraph. We have also elaborated on the advantages of using plumbing rings and medical adhesive – “The application of plumbing rings with medical adhesive to create an apical reservoir on ex vivo corneas does not require extensive technical skill. Moreover, the reagents used are inexpensive and readily available to any lab.”

2. Consider briefly addressing the clinical relevance of the findings, such as potential applications for human ocular infections.

We have now addressed this and added appropriate text to the abstract, introduction and discussion sections.

3. The CFU analysis in Figure 2 provides compelling evidence of moxifloxacin’s efficacy. Could the authors clarify whether the dose-dependent reduction was statistically significant across all tested concentrations?

Statistical analysis was indeed performed and the reduction in CFU was determined to be statistically significant. We have denoted asterisks above the data in Figure 2 and have now included a section detailing statistical analysis.

4. Figure 3 shows the restoration of corneal clarity at 20 µM moxifloxacin. The discussion notes incomplete bacterial clearance despite reduced opacity. Can the authors speculate further on the threshold of bacterial load required to induce opacity?

This is a good and interesting question, albeit difficult to answer. Extent of bacterial load may not directly translate to a proportional increase in opacity. We hypothesise that the extent of opacity stems from the virulence of the bacteria and cannot necessarily be pinpointed to a measure of growth. The bacterial virulence factors modulate the structure of the epithelium and endothelium resulting in loss of translucent properties. In this regard, we hypothesise that high numbers of bacterial CFU with lower pathogenicity would not cause as much opacity as lower bacterial CFU with increased virulence output. We have now included this observation within the results and discussion – “The extent of bacterial load may not directly translate to a proportional increase in corneal opacity. We hypothesise that the degree of opacity is primarily driven by bacterial virulence factors rather than the bacterial load itself. These factors modulate the structure of the epithelium and endothelium, leading to a loss of corneal transparency. As such, high CFU counts with low virulence output may result in less opacity than lower CFU counts with higher virulence.”

5. The histological analysis (Figure 4) effectively demonstrates structural preservation with moxifloxacin treatment. However, additional quantitative metrics, such as epithelial thickness or stromal edema, could strengthen the findings.

We thank the reviewer for their valuable feedback. While the preservation of the epithelium is visually clear, the extent of stromal swelling is less apparent and difficult to assess by eye. To address this, we have incorporated additional quantitative analysis into our study. In the revised manuscript, we used ImageJ to quantify stromal swelling in corneal sections by measuring the percentage of white space within the stroma. The percentage area of white pixels in the stromal region was calculated and subsequently analysed using GraphPad Prism. These results are now presented in Figure 4B.

6. The preparation of bacterial inoculum and porcine corneas is described in detail. However, the use of 70% ethanol for epithelial debridement might alter corneal properties. Have the authors considered alternative methods?

In previous studies, we have employed both the needle scratch and 70% ethanol debridement methods. Whilst the needle scratch method is effective, we also find it to be more technically challenging and thus less reproducible. We highlight in this study that both methods can be used but advocate the latter for consistency. With this, to the manuscript we have added “In this study, we employed ethanol debridement as we found this to be less technically challenging than needle scratching and more reproducible.” As for the question of the potential alteration of corneal properties, we observed by histological analysis that the ethanol debridement method, after incubation, did not result in any obvious morphological changes.

7. The choice of 10 mm plumbing rings is practical, but information on how ring size or adhesive type affects results would be beneficial.

We thank the reviewer for their comment as this is an interesting point to raise, which we can address and improve the quality of the publication. The size of 10 mm is important as it provides a substantial area for inoculum and topical treatments so that analysis outputs can be maximised. 10 mm plumbing rings were consistently able to be fitted to all porcine corneas tested, whereas larger sizes would sometimes become imposed onto the sclera, due to variations in eye sizes obtained from the abattoir. Indeed, smaller size plumbing rings may be used if a smaller initial area of infection is desired, such as for studying infection with herpes. This has now been discussed in the manuscript.

8. The methods lack specific information about the statistical analysis software and parameters. Please provide these details.

Although statistical analysis information is mentioned in the figure legends, we have now provided a section outlining the statistical analysis in the methods (section 2.8).

9. The discussion addresses key limitations, including the absence of immune responses in ex vivo models. Could the authors propose strategies for overcoming these limitations, such as co-culture systems with immune components?

An additional paragraph has been included in section 4.2, where we have discussed co-culture systems, the potential for neutrophil transepithelial migration assays and immune cells, cytokines, and antimicrobial factors using microfluidic platforms.

10. The statement about reduced PcrV expression with moxifloxacin is intriguing but not directly tested in this study. Consider elaborating on how this mechanism aligns with the observed outcomes.

We have expanded results and discussion section 3.2.

11. The discussion could benefit from a stronger conclusion regarding the translational potential of the findings for human therapy.

The conclusion now outlines the clinical translation potential of the developed model. “With this, we present a highly accessible and improved ex vivo infection model, which may lead to better clinical translation of prospective treatments to reduce the burden of MK”.

Minor comments:

1. Abstract: Line 12 “interaction more accurately than” – consider rephrasing for clarity.

The sentence has been re-written for improved clarity.

2. Introduction: Line 43, “P. aeruginosa possesses an array of virulence factors” – consider specifying which are most relevant to MK.

The paragraph has been revised to discuss the roles of Type III secretion system (T3SS) effectors, with a particular focus on ExoU, whose expression is strongly correlated with increased disease severity and poorer clinical outcomes. Additionally, the roles of ExoS and secreted proteases (LasB, protease IV, and PASP) are highlighted as key drivers of MK pathogenesis.

3. Methods: Line 127, “a concentration of 107 CFU/mL” – use superscripts for scientific notation.

This has now been amended.

4. Figures: Ensure figure legends are self-explanatory without requiring text from the main body for context.

The Figure legends have been revised.

5. Formatting: Verify consistent use of italicization for species names throughout the manuscript.

The manuscript has been fully checked for proper italicization.

Responses to Reviewer 2

We thank the reviewer for their encouraging comments on our manuscript. We are pleased that the reviewer recognises the significance of our work in addressing the global challenge of MK and the urgent need for improved models to study this disease. We are also grateful for the acknowledgment of the advantages offered by our ex vivo porcine corneal model, including its simplicity, cost-effectiveness, and ethical acceptability.

We share the reviewer’s perspective on the limitations of the porcine corneal model and have strived to provide a balanced discussion of its strengths and constraints. Once again, we thank the reviewer for their positive feedback.

---

## [Editor Report · Decision Letter 1]

11 Feb 2025

Stable topical application of antimicrobials using plumbing rings in an *ex vivo* porcine corneal infection model

PONE-D-24-57134R1

*Dear Dr. Foulkes,*

*We’re pleased to inform you that your manuscript has been judged scientifically suitable for publication and will be formally accepted for publication once it meets all outstanding technical requirements.*

*Within one week, you’ll receive an e-mail detailing the required amendments. When these have been addressed, you’ll receive a formal acceptance letter and your manuscript will be scheduled for publication.*

*An invoice will be generated when your article is formally accepted. Please note, if your institution has a publishing partnership with PLOS and your article meets the relevant criteria, all or part of your publication costs will be covered. Please make sure your user information is up-to-date by logging into Editorial Manager at Editorial Manager®  and clicking the ‘Update My Information' link at the top of the page. If you have any questions relating to publication charges, please contact our Author Billing department directly at authorbilling@plos.org.*

*If your institution or institutions have a press office, please notify them about your upcoming paper to help maximize its impact. If they’ll be preparing press materials, please inform our press team as soon as possible -- no later than 48 hours after receiving the formal acceptance. Your manuscript will remain under strict press embargo until 2 pm Eastern Time on the date of publication. For more information, please contact onepress@plos.org.*

*Kind regards,*

*Sanhita Roy, Ph.D.*

Academic Editor

*PLOS ONE*

* *

*Additional Editor Comments (optional):*

* *
---

## [Editor Report · Acceptance letter]

PONE-D-24-57134R1

PLOS ONE

Dear Dr. Foulkes,

I'm pleased to inform you that your manuscript has been deemed suitable for publication in PLOS ONE. Congratulations! Your manuscript is now being handed over to our production team.

Kind regards,

on behalf of

Dr. Sanhita Roy

Academic Editor

PLOS ONE